# Equilibrated adaptive learning rates for non-convex optimization

**Yann N. Dauphin**[1]
Université de Montréal
dauphiya@iro.umontreal.ca

**Harm de Vries**[1]
Université de Montréal
devries@iro.umontreal.ca

**Yoshua Bengio**
Université de Montréal
yoshua.bengio@umontreal.ca

## Abstract

Parameter-specific adaptive learning rate methods are computationally efficient ways to reduce the ill-conditioning problems encountered when training large deep networks. Following recent work that strongly suggests that most of the critical points encountered when training such networks are saddle points, we find how considering the presence of negative eigenvalues of the Hessian could help us design better suited adaptive learning rate schemes. We show that the popular Jacobi preconditioner has undesirable behavior in the presence of both positive and negative curvature, and present theoretical and empirical evidence that the so-called equilibration preconditioner is comparatively better suited to non-convex problems. We introduce a novel adaptive learning rate scheme, called ESGD, based on the equilibration preconditioner. Our experiments show that ESGD performs as well or better than RMSProp in terms of convergence speed, always clearly improving over plain stochastic gradient descent.

## 1   Introduction

One of the challenging aspects of deep learning is the optimization of the training criterion over millions of parameters: the difficulty comes from both the size of these neural networks and because the training objective is non-convex in the parameters. Stochastic gradient descent (SGD) has remained the method of choice for most practitioners of neural networks since the 80's, in spite of a rich literature in numerical optimization. Although it is well-known that first-order methods considerably slow down when the objective function is ill-conditioned, it remains unclear how to best exploit second-order structure when training deep networks. Because of the large number of parameters, storing the full Hessian or even a low-rank approximation is not practical, making parameter specific learning rates, i.e diagonal preconditioners, one of the viable alternatives. One of the open questions is how to set the learning rate for SGD adaptively, both over time and for different parameters, and several methods have been proposed (see e.g. Schaul et al. (2013) and references therein).

On the other hand, recent work (Dauphin et al., 2014; Choromanska et al., 2014) has brought theoretical and empirical evidence suggesting that local minima are with high probability not the main obstacle to optimizing large and deep neural networks, contrary to what was previously believed: instead, saddle points are the most prevalent critical points on the optimization path (except when we approach the value of the global minimum). These saddle points can considerably slow down training, mostly because the objective function tends to be ill-conditioned in the neighborhood of

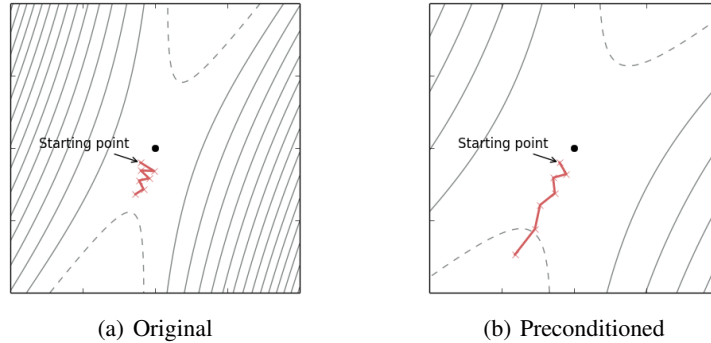

|  (a) Original | (b) Preconditioned |

Figure 1: Contour lines of a saddle point (black point) problem for (a) original function and (b) transformed function (by equilibration preconditioner). Gradient descent slowly escapes the saddle point in (a) because it oscillates along the high positive curvature direction. For the better conditioned function (b) these oscillations are reduced, and gradient descent makes faster progress.

these saddle points. This raises the question: can we take advantage of the saddle structure to design good and computationally efficient preconditioners?

In this paper, we bring these threads together. We first study diagonal preconditioners for saddle point problems, and find that the popular Jacobi preconditioner has unsuitable behavior in the presence of both positive and negative curvature. Instead, we propose to use the so-called equilibration preconditioner and provide new theoretical justifications for its use in Section 4. We provide specific arguments why equilibration is better suited to non-convex optimization problems than the Jacobi preconditioner and empirically demonstrate this for small neural networks in Section 5. Using this new insight, we propose a new adaptive learning rate schedule for SGD, called ESGD, that is based on the equilibration preconditioner. In Section 7 we evaluate the proposed method on two deep autoencoder benchmarks. The results, presented in Section 8, confirm that ESGD performs as well or better than RMSProp. In addition, we empirically find that the update direction of RMSProp is very similar to equilibrated update directions, which might explain its success in training deep neural networks.

## 2 Preconditioning

It is well-known that gradient descent makes slow progress when the curvature of the loss function is very different in separate directions. The negative gradient will be mostly pointing in directions of high curvature, and a small enough learning rate have to be chosen in order to avoid divergence in the largest positive curvature direction. As a consequence, the gradient step makes very little progress in small curvature directions, leading to the slow convergence often observed with first-order methods.

Preconditioning can be thought of as a geometric solution to the problem of pathological curvature. It aims to locally transform the optimization landscape so that its curvature is equal in all directions. This is illustrated in Figure 1 for a two-dimensional saddle point problem using the equilibration preconditioner (Section 4). Gradient descent method slowly escapes the saddle point due to the typical oscillations along the high positive curvature direction. By transforming the function to be more equally curved, it is possible for gradient descent to move much faster.

More formally, we are interested in minimizing a function $f$ with parameters $\theta \in \mathbb{R}^{\mathbf{N}}$. We introduce preconditioning by a linear change of variables $\hat{\theta} = \mathbf{D}^{\frac{1}{2}}\theta$ with a non-singular matrix $\mathbf{D}^{\frac{1}{2}}$. We use this change of variables to define a new function $\hat{f}$, parameterized by $\hat{\theta}$, that is equivalent to the original function $f$:

$$\hat{f}(\hat{\theta}) = f(\mathbf{D}^{-\frac{1}{2}}\hat{\theta}) = f(\theta) \tag{1}$$

The gradient and the Hessian of this new function $\hat{f}$ are (by the chain rule):

$$\nabla \hat{f}(\hat{\theta}) = \mathbf{D}^{-\frac{1}{2}}\nabla f(\theta) \tag{2}$$

$$\nabla^2 \hat{f}(\hat{\theta}) = \mathbf{D}^{-\frac{1}{2}\top}\mathbf{H}\mathbf{D}^{-\frac{1}{2}} \quad \text{with} \quad \mathbf{H} = \nabla^2 f(\theta) \tag{3}$$

A gradient descent iteration $\hat{\theta}_t = \hat{\theta}_{t-1} - \eta \nabla \hat{f}(\hat{\theta})$ for the transformed function corresponds to

$$\theta_t = \theta_{t-1} - \eta \mathbf{D}^{-1} \nabla f(\theta) \tag{4}$$

for the original parameter $\theta$. In other words, by left-multiplying the original gradient with a positive definite matrix $D^{-1}$, we effectively apply gradient descent to the problem after a change of variables $\hat{\theta} = \mathbf{D}^{\frac{1}{2}}\theta$. The curvature of this transformed function is given by the Hessian $\mathbf{D}^{-\frac{1}{2}\top}\mathbf{H}\mathbf{D}^{-\frac{1}{2}}$, and we aim to seek a preconditioning matrix $\mathbf{D}$ such that the new Hessian has equal curvature in all directions. One way to assess the success of $\mathbf{D}$ in doing so is to compute the relative difference between the biggest and smallest curvature direction, which is measured by the condition number of the Hessian:

$$\kappa(\mathbf{H}) = \frac{\sigma_{\max}(\mathbf{H})}{\sigma_{\min}(\mathbf{H})} \tag{5}$$

where $\sigma_{\max}(\mathbf{H}), \sigma_{\min}(\mathbf{H})$ denote respectively the biggest and smallest singular values of $\mathbf{H}$ (which are the absolute value of the eigenvalues). It is important to stress that the condition number is defined for both definite and indefinite matrices.

The famous Newton step corresponds to a change of variables $\mathbf{D}^{\frac{1}{2}} = \mathbf{H}^{\frac{1}{2}}$ which makes the new Hessian perfectly conditioned. However, a change of variables only exists[2] when the Hessian $\mathbf{H}$ is positive semi-definite. This is a problem for non-convex loss surfaces where the Hessian might be indefinite. In fact, recent studies (Dauphin et al., 2014; Choromanska et al., 2014) has shown that saddle points are dominating the optimization landscape of deep neural networks, implying that the Hessian is most likely indefinite. In such a setting, $\mathbf{H}^{-1}$ not a valid preconditioner and applying Newton's step without modification would make you move towards the saddle point. Nevertheless, it is important to realize that the concept of preconditioning extends to non-convex problems, and reducing ill-conditioning around saddle point will often speed up gradient descent.

At this point, it is natural to ask whether there exists a valid preconditioning matrix that always perfectly conditions the new Hessian? The answer is yes, and the corresponding preconditioning matrix is the inverse of the absolute Hessian

$$|\mathbf{H}| = \sum_j |\lambda_j| \mathbf{q}_j \mathbf{q}_j^\top, \tag{6}$$

which is obtained by an eigendecomposition of $\mathbf{H}$ and taking the absolute values of the eigenvalues. See Proposition 1 in Appendix A for a proof that $|\mathbf{H}|^{-1}$ is the only (up to a scalar[3]) symmetric positive definite preconditioning matrix that perfectly reduces the condition number.

Practically, there are several computational drawbacks for using $|\mathbf{H}|^{-1}$ as a preconditioner. Neural networks typically have millions of parameters, rendering it infeasible to store the Hessian ($\mathcal{O}(N^2)$), perform an eigendecomposition ($\mathcal{O}(N^3)$) and invert the matrix ($\mathcal{O}(N^3)$). Except for the eigendecomposition, other full rank preconditioners are facing the same computational issues. We therefore look for more computationally affordable preconditioners while maintaining its efficiency in reducing the condition number of indefinite matrices. In this paper, we focus on diagonal preconditioners which can be stored, inverted and multiplied by a vector in linear time. When diagonal preconditioners are applied in an online optimization setting (i.e. in conjunction with SGD), they are often referred to as adaptive learning rates in the neural network literature.

## 3 Related work

The Jacobi preconditioner is one of the most well-known preconditioners. It is given by the diagonal of the Hessian $\mathbf{D}^{\mathbf{J}} = |\mathrm{diag}(\mathbf{H})|$ where $|\cdot|$ is element-wise absolute value. LeCun et al. (1998) proposes an efficient approximation of the Jacobi preconditioner using the Gauss-Newton matrix. The Gauss-Newton has been shown to approximate the Hessian under certain conditions (Pascanu & Bengio, 2014). The merit of this approach is that it is efficient but it is not clear what is lost by the Gauss-Newton approximation. What's more the Jacobi preconditioner has not be found to be competitive for indefinite matrices (Bradley & Murray, 2011). This will be further explored for neural networks in Section 5.

A recent revival of interest in adaptive learning rates has been started by AdaGrad (Duchi et al., 2011). Adagrad collects information from the gradients across several parameter updates to tune the learning rate. This gives us the diagonal preconditioning matrix $\mathbf{D}^{\mathrm{A}} = (\sum_t \nabla f_{(t)}^2)^{-1/2}$ which relies on the sum of gradients $\nabla f_{(t)}$ at each timestep $t$. Duchi et al. (2011) relies strongly on convexity to justify this method. This makes the application to neural networks difficult from a theoretical perspective. RMSProp (Tieleman & Hinton, 2012) and AdaDelta (Zeiler, 2012) were follow-up methods introduced to be practical adaptive learning methods to train large neural networks. Although RMSProp has been shown to work very well (Schaul et al., 2013), there is not much understanding for its success in practice. Preconditioning might be a good framework to get a better understanding of such adaptive learning rate methods.

## 4   Equilibration

Equilibration is a preconditioning technique developed in the numerical mathematics community (Sluis, 1969). When solving a linear system $\mathbf{A}x = b$ with Gaussian Elimination, significant round-off errors can be introduced when small numbers are added to big numbers (Datta, 2010). To circumvent this issue, it is advised to properly scale the rows of the matrix before starting the elimination process. This step is often referred to as row equilibration, which formally scales the rows of $\mathbf{A}$ to unit magnitude in some $p$-norm. Throughout the following we consider 2-norm. Row equilibration is equivalent to multiplying $\mathbf{A}$ from the left by the matrix $\mathbf{D}_{ii}^{-1} = \frac{1}{\|\mathbf{A}_{i,\cdot}\|_2}$. Instead of solving the original system, we now solve the equivalent left preconditioned system $\hat{\mathbf{A}}x = \hat{b}$ with $\hat{\mathbf{A}} = \mathbf{D}^{-1}\mathbf{A}$ and $\hat{b} = \mathbf{D}_i^{-1}b$.

In this paper, we apply the equilibration preconditioner in the context of large scale non-convex optimization. However, it is not straightforward how to apply the preconditioner. By choosing the preconditioning matrix

$$\mathbf{D}_{ii}^{\mathrm{E}} = \|H_{i,\cdot}\|_2, \tag{7}$$

the Hessian of the transformed function $(\mathbf{D}^{\mathrm{E}})^{-\frac{1}{2}\top}\mathbf{H}(\mathbf{D}^{\mathrm{E}})^{-\frac{1}{2}}$ (see Section 2) does not have equilibrated rows. Nevertheless, its spectrum (i.e. eigenvalues) is equal to the spectrum of the row equilibrated Hessian $(\mathbf{D}^{\mathrm{E}})^{-1}\mathbf{H}$ and column equilibrated Hessian $\mathbf{H}(\mathbf{D}^{\mathrm{E}})^{-1}$. Consequently, if row equilibration succesfully reduces the condition number, then the condition number of the transformed Hessian $(\mathbf{D}^{\mathrm{E}})^{-\frac{1}{2}\top}\mathbf{H}(\mathbf{D}^{\mathrm{E}})^{-\frac{1}{2}}$ will be reduced by the same amount. The proof is given by Proposition 2.

From the above observation, it seems more natural to seek for a diagonal preconditioning matrix $\mathbf{D}$ such that $\mathbf{D}^{-\frac{1}{2}}\mathbf{H}\mathbf{D}^{-\frac{1}{2}}$ is row and column equilibrated. In Bradley & Murray (2011) an iterative stochastic procedure is proposed for finding such matrix. However, we did not find it to work very well in an online optimization setting, and therefore stick to the original equilibration matrix $\mathbf{D}^{\mathrm{E}}$.

Although the original motivation for row equilibration is to prevent round-off errors, our interest is in how well it is able to reduce the condition number. Intuitively, ill-conditioning can be a result of matrix elements that are of completely different order. Scaling the rows to have equal norm could therefore significantly reduce the condition number. Although we are not aware of any proofs that row equilibration improves the condition number, there are theoretical results that motivates its use. In Sluis (1969) it is shown that the condition number of a row equilibrated matrix is at most a factor $\sqrt{N}$ worse than the diagonal preconditioning matrix that optimally reduces the condition number. Note that the bound grows sublinear in the dimension of the matrix, and can be quite loose for the extremely large matrices we consider. In this paper, we provide an alternative justification using the following upper bound on the condition number from Guggenheimer et al. (1995):

$$\kappa(\mathbf{H}) < \frac{2}{|\det \mathbf{H}|} \left( \frac{\|\mathbf{H}\|_F}{\sqrt{N}} \right)^N \tag{8}$$

The proof in Guggenheimer et al. (1995) provides useful insight when we expect a tight upper bound to be tight: if all singular values, except for the smallest, are roughly equal.

We prove by Proposition 4 that row equilibration improves this upper bound by a factor $\det(\mathbf{D}^E) \left( \frac{\|\mathbf{H}\|_F}{\sqrt{N}} \right)^N$. It is easy see that the bound is more reduced when the norms of the rows

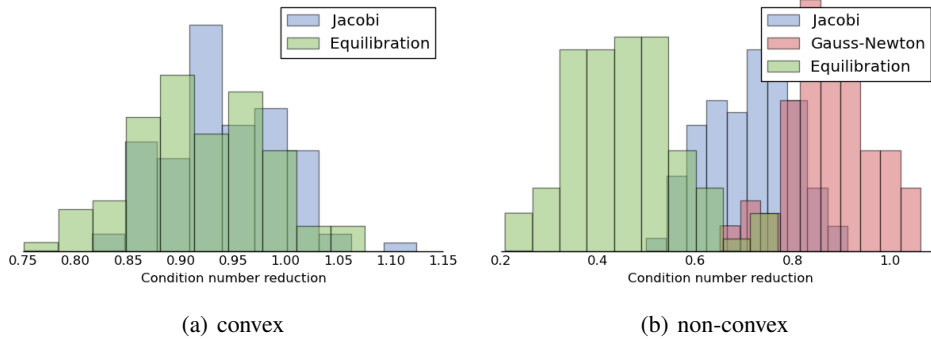

(a) convex                    (b) non-convex

Figure 2: Histogram of the condition number reduction (lower is better) for random Hessians in a (a) convex and b) non-convex setting. Equilibration clearly outperforms the other methods in the non-convex case.

are more varied. Note that the proof can be easily extended to column equilibration, and row and column equilibration. In contrast, we can not prove that the Jacobi preconditioner improves the upper bound, which provides another justification for using the equilibration preconditioner.

A deterministic implementation to calculate the 2-norm of all matrix rows needs to access all matrix elements. This is prohibitive for very large Hessian's that can not even be stored. We therefore resort to a matrix-free estimator of the equilibration matrix that only uses matrix vector multiplications of the form $(\mathbf{H}\mathbf{v})^2$ where the square is element-wise and $\mathbf{v}_i \sim \mathcal{N}(0,1)$[4]. As shown by Bradley & Murray (2011), this estimator is unbiased, i.e.

$$\|\mathbf{H}_{i,\cdot}\|^2 = \mathrm{E}[(\mathbf{H}\mathbf{v})^2]. \qquad (9)$$

Since multiplying the Hessian by a vector can be efficiently done without ever computing the Hessian, this method can be efficiently used in the context of neural networks using the R-operator Schraudolph (2002). The R-operator computation only uses gradient-like computations and costs about the same as two backpropagations.

## 5 Equilibrated learning rates are well suited to non-convex problems

In this section, we demonstrate that equilibrated learning rates are well suited to non-convex optimization, particularly compared to the Jacobi preconditioner. First, the diagonal equilibration matrix can be seen as an approximation to diagonal of the absolute Hessian. Reformulating the equilibration matrix as

$$\mathbf{D}_{ii}^{\mathrm{E}} = \|\mathbf{H}_{i,\cdot}\|_2 = \sqrt{\mathrm{diag}(\mathbf{H}^2)}_i \qquad (10)$$

reveals an interesting connection. Changing the order of the square root and diagonal would give us the diagonal of $|\mathbf{H}|$. In other words, the equilibration preconditioner can be thought of as the Jacobi preconditioner of the absolute Hessian.

Recall that the inverse of the absolute Hessian $|\mathbf{H}|^{-1}$ is the only symmetric positive definite matrix that reduces the condition number to 1 (the proof of which can be be found in Proposition 1 in the Appendix). It can be considered as the gold standard, if we do not take computational costs into account. For indefinite matrices, the diagonal of the Hessian $\mathbf{H}$ and the diagonal of the absolute Hessian $|\mathbf{H}|$ will be very different, and therefore the behavior of the Jacobi and equilibration preconditioner will also be very different.

In fact, we argue that the Jacobi preconditioner can cause divergence because it underestimates curvature. We can measure the amount of curvature in a given direction with the Raleigh quotient

$$R(\mathbf{H}, \mathbf{v}) = \frac{\mathbf{v}^T \mathbf{H} \mathbf{v}}{\mathbf{v}^T \mathbf{v}}. \qquad (11)$$

**Algorithm 1** Equilibrated Gradient Descent

---

**Require:** Function $f(\theta)$ to minimize, learning rate $\epsilon$ and damping factor $\lambda$
   $\mathbf{D} \leftarrow 0$
   **for** $i = 1 \rightarrow K$ **do**
      $\mathbf{v} \sim \mathcal{N}(0, 1)$
      $\mathbf{D} \leftarrow \mathbf{D} + (\mathbf{H}\mathbf{v})^2$
      $\theta \leftarrow \theta - \epsilon \frac{\nabla f(\theta)}{\sqrt{\mathbf{D}/i} + \lambda}$
   **end for**

---

This quotient is large when there is a lot of curvature in the direction $\mathbf{v}$. The Raleigh quotient can be decomposed into $R(\mathbf{H}, \mathbf{v}) = \sum_j^N \lambda_j \mathbf{v}^\top \mathbf{q}_j \mathbf{q}_j^\top \mathbf{v}$ where $\lambda_j$ and $\mathbf{q}_j$ are the eigenvalues and eigenvectors of $\mathbf{H}$. It is easy to show that each element of the Jacobi matrix is given by $\mathbf{D}_{ii}^{\mathrm{J}} = |R(\mathbf{H}, \mathbf{I}_{.,i})|^{-1} = |\sum_j^N \lambda_j \mathbf{q}_{j,i}^2|^{-1}$. An element $\mathbf{D}_{ii}^{\mathrm{J}}$ is the inverse of the sum of the eigenvalues $\lambda_j$. Negative eigenvalues will reduce the total sum and make the step much larger than it should. Specifically, imagine a diagonal element where there are large positive and negative curvature eigendirections. The contributions of these directions will cancel each other and a large step will be taken in that direction. However, the function will probably also change fast in that direction (because of the high curvature), and the step is too large for the local quadratic approximation we have considered.

Equilibration methods never diverge this way because they will not underestimate curvature. In equilibration, the curvature information is given by the Raleigh quotient of the squared Hessian $\mathbf{D}_{ii}^{\mathrm{E}} = (R(\mathbf{H}^2, \mathbf{I}_{.,i}))^{-1/2} = (\sum_j \lambda_j^2 \mathbf{q}_{j,i}^2)^{-1/2}$. Note that all the elements are positive and so will not cancel. Jensen's inequality then gives us an upper bound

$$\mathbf{D}_{ii}^{\mathrm{E}} \leq |\mathbf{H}|_{ii}^{-1}. \tag{12}$$

which ensures that equilibrated adaptive learning rate will in fact be more conservative than the Jacobi preconditioner of the absolute Hessian (see Proposition 2 for proof).

This strengthens the links between equilibration and the absolute Hessian and may explain why equilibration has been found to work well for indefinite matrices Bradley & Murray (2011). We have verified this claim experimentally for random neural networks. The neural networks have 1 hidden layer of a 100 sigmoid units with zero mean unit-variance Gaussian distributed inputs, weights and biases. The output layer is a softmax with the target generated randomly. We also give results for similarly sampled logistic regressions. We compare reductions of the condition number between the methods. Figure 2 gives the histograms of the condition number reductions. We obtained these graphs by sampling a hundred networks and computing the ratio of the condition number before and after preconditioning. On the left we have the convex case, and on the right the non-convex case. We clearly observe that the Jacobi and equilibration method are closely matched for the convex case. However, in the non-convex case equilibration significantly outperforms the other methods. Note that the poor performance of the Gauss-Newton diagonal only means that its success in optimization is not due to preconditioning. As we will see in Section 8 these results extend to practical high-dimensional problems.

## 6 Implementation

We propose to build a scalable algorithm for preconditioning neural networks using equilibration. This method will estimate the same curvature information $\sqrt{\mathrm{diag}(\mathbf{H}^2)}$ with the unbiased estimator described in Equation 9. It is prohibitive to compute the full expectation at each learning step. Instead we will simply update our running average at each learning step much like RMSProp. The pseudo-code is given in Algorithm 1. The additional costs are one product with the Hessian, which is roughly the cost of two additional gradient calculations, and the sampling a random Gaussian vector. In practice we greatly amortize the cost by only performing the update every 20 iterations. This brings the cost of equilibration very close to that of regular SGD. The only added hyper-parameter is the damping $\lambda$. We find that a good setting for that hyper-parameter is $\lambda = 10^{-4}$ and it is robust over the tasks we considered.

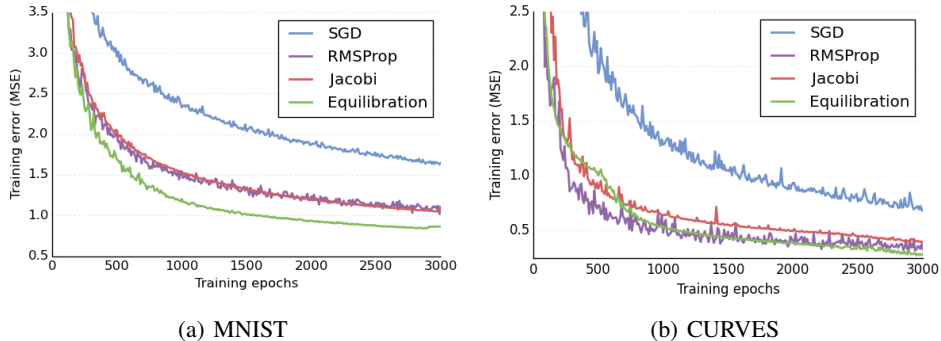

(a) MNIST                                          (b) CURVES

Figure 3: Learning curves for deep auto-encoders on a) MNIST and b) CURVES comparing the different preconditioned SGD methods.

In the interest of comparison, we will evaluate SGD preconditioned with the Jacobi preconditioner. This will allow us to verify the claims that the equilibration preconditioner is better suited for non-convex problems. Bekas et al. (2007) show that the diagonal of a matrix can be recovered by the expression

$$\text{diag}(\mathbf{H}) = \text{E}[\mathbf{v} \odot \mathbf{H}\mathbf{v}] \qquad (13)$$

where $\mathbf{v}$ are random vectors with entries $\pm 1$ and $\odot$ is the element-wise product. We use this estimator to precondition SGD in the same fashion as that described in Algorithm 1. The variance of this estimator for an element $i$ is $\sum_j H_{ji}^2 - H_{ii}^2$, while the method in Martens et al. (2012) has $H_{ii}^2$. Therefore, the optimal method depends on the situation. The computational complexity is the same as ESGD.

# 7   Experimental setup

We consider the challenging optimization benchmark of training very deep neural networks. Following Martens (2010); Sutskever et al. (2013); Vinyals & Povey (2011), we train deep auto-encoders which have to reconstruct their input under the constraint that one layer is very low-dimensional. The networks have up to 11 layers of sigmoidal hidden units and have on the order of a million parameters. We use the standard network architectures described in Martens (2010) for the MNIST and CURVES dataset. Both of these datasets have 784 input dimensions and 60,000 and 20,000 examples respectively.

We tune the hyper-parameters of the optimization methods with random search. We have sampled the learning rate from a logarithmic scale between $[0.1, 0.01]$ for stochastic gradient descent (SGD) and equilibrated SGD (ESGD). The learning rate for RMSProp and the Jacobi preconditioner are sampled from $[0.001, 0.0001]$. The damping factor $\lambda$ used before dividing the gradient is taken from either $\{10^{-4}, 10^{-5}, 10^{-6}\}$ while the exponential decay rate of RMSProp is taken from either $\{0.9, 0.95\}$. The networks are initialized using the sparse initialization described in Martens (2010). The minibatch size for all methods in 200. We do not make use of momentum in these experiments in order to evaluate the strength of each preconditioning method on its own. Similarly we do not use any regularization because we are only concerned with optimization performance. For these reasons, we report training error in our graphs. The networks and algorithms were implemented using Theano Bastien et al. (2012), simplifying the use of the R-operator in Jacobi and equilibrated SGD. All experiments were run on GPU's.

# 8   Results

## 8.1   Comparison of preconditioned SGD methods

We compare the different adaptive learning rates for training deep auto-encoders in Figure 3. We don't use momentum to better isolate the performance of each method. We believe this is important because RMSProp has been found not to mix well with momentum (Tieleman & Hinton, 2012). Thus the results presented are not state-of-the-art, but they do reach state of the art when momentum is used.

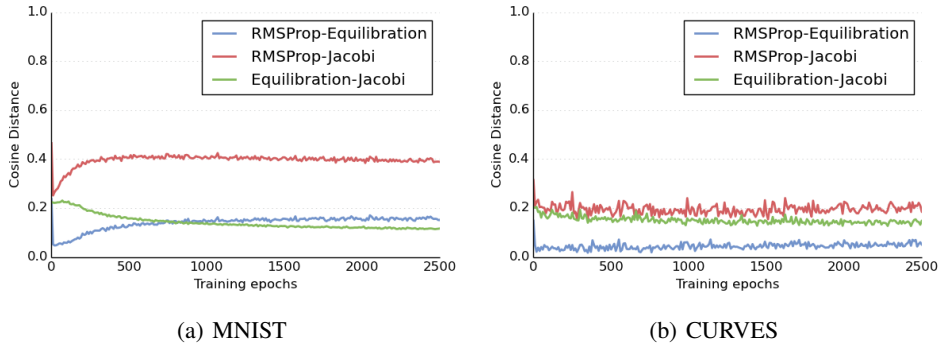

|                | (a) MNIST | (b) CURVES |

Figure 4: Cosine distance between the diagonals estimated by each method during the training of a deep auto-encoder trained on a) MNIST and b) CURVES. We can see that RMSProp estimates a quantity close to the equilibration matrix.

Our results on MNIST show that the proposed ESGD method significantly outperforms both RMSProp and Jacobi SGD. The difference in performance becomes especially notable after 250 epochs. Sutskever et al. (2013) reported a performance of 2.1 of training MSE for SGD without momentum and we can see all adaptive learning rates improve on this result, with equilibration reaching 0.86. We observe a convergence speed that is approximately three times faster then our baseline SGD. ESGD also performs best for CURVES, although the difference with RMSProp and Jacobi SGD is not as significant as for MNIST. We show in the next section that the smaller gap in performance is due to the different preconditioners behaving the same way on this dataset.

## 8.2 Measuring the similarity of the methods

We train deep autoencoders with RMSProp and measure every 10 epochs the equilibration matrix $\mathbf{D}^E = \sqrt{\text{diag}(\mathbf{H}^2)}$ and Jacobi matrix $\mathbf{D}^J = \sqrt{\text{diag}(\mathbf{H})^2}$ using 100 samples of the unbiased estimators described in Equations 9, respectively. We then measure the pairwise differences between these quantities in terms of the cosine distance $\text{cosine}(u, v) = 1 - \frac{u \cdot v}{\|u\| \|v\|}$, which measures the angle between two vectors and ignores their norms.

Figure 4 shows the resulting cosine distances over training on MNIST and CURVES. For the latter dataset we observe that RMSProp remains remarkably close (around 0.05) to equilibration, while it is significantly different from Jacobi (in the order of 0.2). The same order of difference is observed when we compare equilibration and Jacobi, confirming the observations of Section 5 that both quantities are rather different in practice. For the MNIST dataset we see that RMSProp fairly well estimates $\sqrt{\text{diag}(\mathbf{H})^2}$ in the beginning of training, but then quickly diverges. After 1000 epochs this difference has exceeded the difference between Jacobi and equilibration, and RMSProp no longer matches equilibration. Interestingly, at the same time that RMSProp starts diverging, we observe in Figure 3 that also the performance of the optimizer drops in comparison to ESGD. This may suggests that the success of RMSProp as a optimizer is tied to its similarity to the equilibration matrix.

## 9  Conclusion

We have studied diagonal preconditioners for saddle point problems i.e. indefinite matrices. We have shown by theoretical and empirical arguments that the equilibration preconditioner is comparatively better suited to this kind of problems than the Jacobi preconditioner. Using this insight, we have proposed a novel adaptive learning rate schedule for non-convex optimization problems, called ESGD, which empirically outperformed RMSProp on two competitive deep autoencoder benchmark. Interestingly, we have found that the update direction of RMSProp was in practice very similar to the equilibrated update direction, which might provide more insight into why RMSProp has been so successfull in training deep neural networks. More research is required to confirm these results. However, we hope that our findings will contribute to a better understanding of SGD's adaptive learning rate schedule for large scale, non-convex optimization problems.

## Footnotes

[1]Denotes first authors

[2] A real square root $\mathbf{H}^{\frac{1}{2}}$ only exists when $\mathbf{H}$ is positive semi-definite.

[3] can be incorporated into the learning rate

[4]Any random variable $\mathbf{v}_i$ with zero mean and unit variance can be used.

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
