[Reviews · NeurIPS 2015]

Submitted by Assigned_Reviewer_1

Paper 921 introduces a new variant of the adaptive SGD method for non-convex optimization called "equilibrated gradient descent". The paper states that the equilibration preconditioner is better than Jacobi preconditioner theoretically and empirically.

For the experiments, MNIST and CURVES are two easy tasks and it cannot say too much insight about the proposed algorithm. I think more experiments should be implemented. When we use neural network, what we really cares are the test accuracies rather than "training error (MSE)". On the other hand, I think it is not enough to compare the ESGD with SGD when the x-axis is "epoch".

There should be another figure whose x-axis is the training time measured by "minute" or "hour".

The paper is not easy to understand clearly. (1) In Algorithm 1, should $H$ be re-calculated every iteration? If yes, there should be a subscript for $H$. Also, in line 277 (the update of $\theta$), should it be $\sqrt{D / i}$? (2) I think in Section 4, paper 921 tries to state that it is the $D^{E}$ that can greatly reduce the condition number $\kappa(D^{E-1}H)$ and I think it should be declared clearly. (3) In line 178, what does $D^{-1}=\Vert A_{I,.}\Vert_2$ mean? (4) In line 267, $R(H,v)=...$ is a vector while (11) is a scalar, so what do you want to express here? Since there is no theorem that can provide a convergence rate for the proposed ESGD, I think a clearer explanation is needed.

Overall, I expect more convincing experiments and a clearer explanation of the proposed ESGD.
Summary: Overall, I expect more convincing experiments and a clearer explanation of the proposed ESGD.

Submitted by Assigned_Reviewer_2

The authors describe the approach of pre-conditioning, a numerical method that uses a linear change of variables to perform gradient descent in a better conditioned parameter space. The authors propose using a pre-conditioner known as "equilibration", which row-normalizes a matrix, to pre-condition the loss Hessian. They then propose a practical numerical technique to estimate the equlibration pre-conditioner, which costs little more than normal gradient-based training. They show experimentally that this approach reduces the condition number of random Hessians, and they show theoretically that it reduces an upper-bound on the condition number of the Hessian, though a more direct proof is lacking. They further show that the resulting algorithm "Equilibrated Gradient Descent" (EGD) improves the convergence time of deep neural networks on some data sets. Additionally, they argue that the heuristic algorithm RMSProp is an approximation to equilibrated EGD.

I think this is a clear, insightful paper with a well-rounded analysis. It is original and may be immediately useful for neural network training.
Summary: This is a well-written paper with clear and precise experimental results and good theory on a topic of reasonable importance -- a justification of the success of one of the most common online training algorithms for neural networks and also the provision of a new, fairly practical online training algorithm.

Submitted by Assigned_Reviewer_3

This paper proposes a new adaptive learning rate scheme for optimizing nonlinear objective functions that arise during the training of deep neural networks. The main argument is based on recent results that indicate that the difficulty of the optimization stems from the presence of saddle points rather than local minima in the optimization path.

The saddle points slow down training since the objective function tends to be flat in many directions and ill-conditioned in the neighbourhood of the saddle points. The authors propose a new method for reducing the ill-conditioning (the problem of pathological curvature) by "preconditioning" the objective function through a linear change of variables, which reduces to left-multiplying the gradient descent update step with a learned preconditioning matrix D. They focus specifically on the case where D is diagonal, and they show how a diagonal D reduces to methods for learning parameter-specific learning rates, such as the well-known Jacobi preconditioner or RMSProp.

This is a nice framework within which to consider different schemes for adaptive learning rates. As a side note, it was unclear to me whether this link has been established before, or whether this was a contribution of this paper?

Their main *theoretical* contribution is to show that the Jacobi preconditioner has undesirable behaviour in the presence of both positive and negative curvature, and to derive the optimal preconditioner which they call the "equilibration preconditioner" D^E as |H|^-1. However, since D^E is a function of the Hession which is O(n^2) in the number of weights of the network, both computing it and storing it becomes computationally intractable. Their main *practical* contribution is an efficient approximate algorithm "Equilibrated Stochastic Gradient Descent" (ESGD) which estimates the optimal D^E by using the R-operator and requiring roughly two additional gradient calculations (further amortized by only performing this step every ~20 iterations) and sampling a vector from a Gaussian.

The theoretical and practical contributions are verified by their empirical results which show that ESGD outperforms both standard SGD, Jacobi preconditioning and RMSProp for training deep auto-encoder networks.

Overall, the paper is fairly clearly written, although I had some issues with ambiguous notation, primarily between distinguishing between element-wise operators and matrix-operators and at some points where it appears that matrices are defined element-wise but equated with full matrices, or vice versa, e.g.:

* Eqn 7 and the definition for D^-1 = 1 / ||A_{i, .}||_2 just above it. Do you mean here the i'th element of D^-1 is defined like that? Otherwise, what does the subscript i refer to in A_{i,.} ? * Eqn 10 (which I'm taking to mean that each element i on the diagonal of D^E is defined as the norm of the i'th row of the Hessian, ||H_{i,.}||, is that correct?).

Overall it would be helpful to add a section at the beginning on your notation, and specifying which operators act element-wise (e.g. |A|) and how you index into matrices, e.g. what do you mean by

* A_{i,.} vs A_i (I'm presuming one indexes a row from a full matrix and the other an element from a diagonal matrix?), or

* q_i q_j (e.g. below Eqn 11) vs q_{j,i} (in the supplementary material). Are these both the same?

In the supplementary material, it was not clear to me in Proposition 3 how the q on the RHS of the inequality is still squared after applying (q_*^2)^-{1/2} which seems to be q_*^{-1}?
Summary: This is a well-written paper that presents a new adaptive learning rate scheme for training deep neural networks. It establishes theoretical links between different methods within a preconditioning framework, derives and presents an efficient approximation to the optimal preconditioner and gives empirical results showing that it works better than three alternatives in practice.

Author Feedback
Author rebuttal: Thanks for all the noted typos and required clarifications, which will be dealt with in the revision.

Assigned_Reviewer_2
-------------------
We will clarify the equations in question.
- In general, we indeed denote the norm of a row by A_{i, \cdot}, while D_i is the i-th diagonal element of a diagonal matrix D.
- q_j indicates the j-th eigenvector of the Hessian matrix, while q_{j, i} means the i-th element of the j-th eigenvector of H. We will clarify this notation in the paper.
- In Eq 7 and just above it, it is indeed the expression of the i-th element and it should have been D^E_i = .. and D^{-1}_i.
- Your interpretation of Eq 10 is correct (see first bullet point). The notation should have been D^E_i = \|H_{i, \cdot}\|_i = \sqrt{diag(H^2)}_i
- Proposition 3 is the result of applying the finite form of Jensen's inequality with q^2 as the positive weights. It is due to this that the square-root is only applied to the eigenvalues in this bound.

Assigned_Reviewer_4
-------------------
We have used the tasks that were established in previous research on optimizing neural networks. Such as
(Martens, 2010) http://www.cs.toronto.edu/~jmartens/docs/Deep_HessianFree.pdf,
(Vinyals et al, 2011) http://arxiv.org/pdf/1111.4259v1.pdf and
(Sutskever et al, 2013) http://www.cs.toronto.edu/~fritz/absps/momentum.pdf.

We show training error because the goal of this paper is to demonstrate improved optimization performance.
Although generalization error is of course what we care about at the end, it compounds other factors such as
regularization effects and the relative amount of training data available (compared to the flexibility
of the architecture), potentially obscuring optimization improvements (which can lead to overfitting
if there are not enough regularizers, too few examples, too many parameters, etc.).

Thanks for the suggestion to compare the optimization algorithms in terms of training time
rather than number of epochs. We will include those plots in the next version.

We will clarify the notation.
- Products must be computed with the Hessian at the current step.
- On line 277, it should indeed be i. (And the for loop should run from i = 1 -> K)
- On line 178, the value computed is the norm of each row.
- On line 267, they should both be scalars. The typo is a missing product with v i.e. \sum_j \lambda_j v^T q_j q_j^T v. (Furthermore, we assume that the direction v in which we measure the curvature is normalized such that v^T v = 1)

Assigned_Reviewer_5
-------------------
We did not include momentum in the evaluation in order to isolate the effect of the preconditioner.
In future studies we would like to investigate whether preconditoning and momentum can be effectively combined.

Assigned_Reviewer_6
-------------------
(Sutskever et al, 2013) http://www.cs.toronto.edu/~fritz/absps/momentum.pdf found that a batch size of 200
for SGD was suitable for these tasks. We also use that batch size in our experiments for ESGD.
ESGD is less affected by the batch size than it would otherwise would be due to the use of the running average.